# Lateral Export of Dissolved Inorganic and Organic Carbon from a Small Mangrove Estuary with Tidal Fluctuation

**Toshiyuki Ohtsuka [1],\*, Takeo Onishi [2], Shinpei Yoshitake [3], Mitsutoshi Tomotsune [4], Morimaru Kida [5], Yasuo Iimura [6], Miyuki Kondo [7], Vilanee Suchewaboripont [8], Ruoming Cao [1], Kazutoshi Kinjo [9] and Nobuhide Fujitake [10]**

[1]  River Basin Research Centre (RBRC), Gifu University, 1-1 Yanagito, Gifu-City 501-1193, Gifu, Japan; caoruoming@gmail.com

[2]  Faculty of Applied Biological Sciences, Gifu University, 1-1 Yanagito, Gifu-City 501-1193, Gifu, Japan; takeon@gifu-u.ac.jp

[3]  Faculty of Education and Integrated Arts and Sciences, Waseda University, 2-2 Wakamatsucho, Shinjuku-ku, Tokyo 162-8480, Japan; syoshi@waseda.jp

[4]  College of Agriculture, Tamagawa University, 6-1-1 Tamagawagakuen, Machida, Tokyo 194-8610, Japan; tomo.32104@agr.tamagawa.ac.jp

[5]  Research Group for Marine Geochemistry (ICBM-MPI Bridging Group), Institute for Chemistry and Biology of the Marine Environment (ICBM), University of Oldenburg, Carl-von-Ossietzky-Str. 9-11, 26129 Oldenburg, Germany; morimaru.kida@uol.de

[6]  School of Environmental Sciences, The University of Shiga Prefecture, 2500 Hassaka-cho, Hikone-City 522-8583, Shiga, Japan; iimura.y@ses.usp.ac.jp

[7]  Center for Environmental Measurement and Analysis, National Institute for Environmental Studies (NIES), 16-2 Onogawa, Tsukuba 305-8506, Ibaraki, Japan; kondo.miyuki@nies.go.jp

[8]  The Institute for the Promotion of Teaching and Technology, Khlong Toei, Bangkok 10110, Thailand; vsuch@ipst.ac.th

[9]  Faculty of Agriculture, University of the Ryukyu, 1 Senbaru, Nishihara, Nakagami, Okinawa 903-0213, Japan; wa614@agr.u-ryukyu.ac.jp

[10]  Graduate School of Agricultural Science, Kobe University, 1-1 Rokkodai-cho, Nada-ku, Kobe 657-8501, Japan; fujitake@kobe-u.ac.jp

\*  Correspondence: ohtsuka@gifu-u.ac.jp

**Abstract:** The significance of aquatic lateral carbon (C) export in mangrove ecosystems highlights the extensive contribution of aquatic pathways to the net ecosystem carbon budget. However, few studies have investigated lateral fluxes of dissolved organic carbon (DOC) and inorganic carbon (DIC), partly due to methodological difficulty. Therefore, we evaluated area-based lateral C fluxes in a small mangrove estuary that only had one exit for water exchange to the coast. We sampled water from the mouth of the creek and integrated discharge and consecutive concentration of mangrove-derived C ($\Delta$C). Then, we estimated the area-normalized C fluxes based on the inundated mangrove area. DIC and DOC concentrations at the river mouth increased during ebb tide during both summer and winter. We quantified the $\Delta$C in the estuary using a two-component conservative mixing model of freshwater and seawater. DIC and DOC proportions of $\Delta$C concentrations at the river mouth during ebb tide was between 34% and 56% in the winter and 26% and 42% in the summer, respectively. DIC and DOC fluxes from the estuary were estimated to be 1.36 g C m$^{-2}$ d$^{-1}$ and 0.20 g C m$^{-2}$ d$^{-1}$ in the winter and 3.35 g C m$^{-2}$ d$^{-1}$ and 0.86 g C m$^{-2}$ d$^{-1}$ in the summer, respectively. Based on our method, daily fluxes are mangrove area-based DIC and DOC lateral exports that can be directly incorporated into the mangrove carbon budget.

**Keywords:** carbon budget; carbon cycling; dissolved inorganic carbon; dissolved organic carbon; lateral carbon flux; mangrove forest; water flow; $\delta^{13}$C-DIC

## 1. Introduction

Mangrove forests store a vast amount of "blue carbon", which contributes to 10–15% of carbon burial in global coastal sediments, although they occupy only 0.5% of the world's coastal zone [1] and have been recognized as the most carbon-rich ecosystem in the world [2]. Global potential $CO_2$ emissions from mangrove loss are estimated to be ~7.0 Tg $CO_2$ year$^{-1}$, and thus, mangrove conservation may be a low-cost method of reducing $CO_2$ emission [3]. The huge carbon sequestration abilities of mangrove forests may result from being highly productive ecosystems with low heterotrophic respiration (Rh) under submerged anaerobic soil, i.e., mangrove forests have high net ecosystem production (NEP) [4]. For example, Barr et al. [5] revealed unusually high NEP (11.7 ± 1.27 Mg C ha$^{-1}$ year$^{-1}$) in a mangrove forest of Everglades National Park. They concluded that the high NEP was due to relatively low respiration rates, which were more similar to temperate forests than tropical forests. Poungparn et al. [6] also suggested that high net primary productivity (NPP, 9.35 to 12.9 Mg C ha$^{-1}$ year$^{-1}$) and low Rh (from 1.72 to 2.63 Mg C ha$^{-1}$ year$^{-1}$) increased the NEP (up to 11.3 Mg C ha$^{-1}$ year$^{-1}$) of a secondary mangrove forest in eastern Thailand. These NEP values of mangrove forests are more than twofold greater than those of upland forests in East Asia [7].

Alternatively, the traditional "outwelling hypothesis" suggests that a significant fraction of the organic matter produced by mangrove trees is exported to the coastal ocean [8]. Considerable research efforts have been directed toward studying lateral organic carbon (C) flux from mangrove forest with tide activity. Adame and Lovelock [9] reviewed the carbon exchange of mangrove forests, including dissolved organic C (DOC), particulate organic C (POC), and C flux as litter. The mean lateral C fluxes of DOC, POC, and litter from mangrove forests were −26.6 ± 88.0, −59.1 ± 88.0, and −202.0 ± 158.0 g C m$^{-2}$ year$^{-1}$, respectively (negative values denote exports from the mangrove). These data were not the estimation of solely mangrove-derived C export but of net C exchange (inwellings-outwellings) that may include contributions from external systems, such as terrestrial, riverine, and oceanic systems. Bouillon et al. [10] also revealed that benthic mineralization as a part of Rh and subsequent export as dissolved inorganic C (DIC) could represent a very significant and unaccounted flux of mangrove-derived C. Recent studies (e.g., [11–13]) confirmed that most of the carbon being tidally exported from mangrove forest is DIC, a result of organic matter mineralization and porewater input from mangrove ecosystems. These studies suggest that DIC outwelling as the part of the Rh appears to be a common phenomenon in mangrove forests and that previous studies of mangrove NEP, which did not consider the potential impact of tidal effect, may greatly overestimate the NEP (e.g., [5,6,14]). Therefore, more empirical work on carbon processes is required to elucidate where and how mangrove forests sequester blue carbon [15,16].

The study of the origin and quantity of DIC (and DOC) in mangrove-fringed estuaries is increasing from the viewpoint of the oceanic environment. For example, Miyajima et al. [17] quantitatively estimated the DIC exported from an estuary in Southeast Asian mangrove forest using isotope mass-balancing models for $\delta^{13}$C-DIC. They revealed that $^{13}$C-depleted DIC inputs from the riverside mangrove and the concentration of mangrove-derived DIC at the estuary were estimated to be as high as 856 μmol L$^{-1}$. Although aquatic lateral C fluxes may offset huge NEP values, there are few studies concerning mangrove C budgets, including aquatic lateral C fluxes, partly due to the difficulty in constraining mangrove area-based estimates of lateral C fluxes. Sippo et al. [18] calculated the net exchange of DIC in six pristine mangrove tidal creeks and normalized the exchange of DIC to the catchment area as the net rate of DIC export as 59 mmol m$^{-2}$ d$^{-1}$ (approximately 2.6 Mg C ha$^{-1}$ year$^{-1}$). However, the watershed-based estimate is difficult to scale down to mangrove area-based DIC flux because mangrove forests did not cover the entire watershed, and tidal amplitude (i.e., discharge from

the mangrove forests) and species composition significantly differ depending on the spatial variations of the mangrove forests. We need to know mangrove area-based estimates of not only Rh (including $CO_2$ flux and DIC export) but also organic carbon lateral fluxes (DOC, POC, and litter) to understand C processes in mangrove ecosystems.

Ishigaki Island possesses the second largest mangrove area in southwestern subtropical Japan. The Fukido River Basin in Ishigaki Island has a small watershed and a well-defined small mangrove creek around the river mouth [19,20], which is usually inundated with seawater twice a day. This kind of small estuarine type of mangrove with only one entrance (river mouth) of water flow should serve as a good model to study the whole system area-based carbon budget. In 2014, we demarcated a large permanent plot (0.64 ha) in the mangrove estuary and reported that Rh (soil surface $CO_2$ flux) was very low (2.6–2.9 Mg C ha$^{-1}$ year$^{-1}$) [21], while aboveground NPP was fairly high (5.3 Mg C ha$^{-1}$ year$^{-1}$) compared with the Rh [19]. We hypothesized that a small estuarine mangrove that has one entrance of water flow is ideal for estimating lateral carbon fluxes with the tidal fluctuation. To test this hypothesis, we measured consecutive dissolved C concentration and discharge water flux at the river mouth. The objectives of this paper were 1) to evaluate the daily change of DIC and DOC concentration at the river mouth at 1 h resolution in winter and summer and 2) to estimate mangrove-derived DIC and DOC flux with the tidal fluctuation in the small estuarine mangrove in southwestern Japan.

## 2. Materials and Methods

### 2.1. Study Site

The study site is situated in the estuary of Fukido River on Ishigaki Island, southwestern Japan (Figure 1a). The Fukido River is short (<1.5 km), and the area of the watershed is 2.57 km$^2$, including the mangrove estuary (19 ha), which occurs around the river mouth. Natural subtropical evergreen broad-leaved forests (or lucidophyllous forests) occupy ~95% of the watershed, with the rest of land use being sparse sugarcane and paddy fields (0.11 km$^2$) [22]. There are only two mangrove species in the estuary: *Bruguiera gymnorrhiza* and *Rhizophora stylosa*. Tree densities of *B. gymnorrhiza* and *R. stylosa* were 2323 ha$^{-1}$ and 594 ha$^{-1}$, respectively, in the permanent plot [19]. The aboveground biomass was considerably high at 164.6 Mg d.m. (dry matter) ha$^{-1}$ and was occupied 84% by *B. gymnorrhiza* [19]. The aboveground NPP of the mangrove from 2014 to 2016 was estimated to be 10.66 ± 1.46 Mg d.m. ha$^{-1}$ year$^{-1}$, which included woody increment (3.10 ± 0.51 Mg d.m. ha$^{-1}$ year$^{-1}$) and litter production (7.56 ± 0.99 Mg d.m. ha$^{-1}$ year$^{-1}$) [19]. The proportion of *B. gymnorrhiza* was more than 85% of woody increment, and the mangrove forest was dominated by *B. gymnorrihza* in a mature late-successional forest [19]. The forest soil was mineral rather than peaty (60–70% sand content) without a litter layer and had a depth of approximately 1 m before the limestone bedrock [23]. The tentative soil type was gley soil, and the soil organic carbon content did not change systematically with depth and varied between 2% and 7% [23,24]. The soil carbon pool in the mangrove forest was estimated to be 261.5 ± 53.2 Mg C ha$^{-1}$ [24].

The study area has a subtropical monsoon climate. The average annual precipitation (1981–2000) was 2169 mm and was distributed throughout the year, although the precipitation in August to September is relatively large, mainly due to typhoons (Figure 1b). The annual mean temperature was 23.8 °C, and the mean temperatures in the coldest month (January) and in the hottest month (July) were 18.3 °C and 28.8 °C, respectively (Figure 1b). The estuary has a clear semidiurnal tide (two tidal cycles per day) and is heavily influenced by seawater. The tidal height ranges between 0.5 and 2 m at low and high tides, respectively, at the river mouth. Salinity at high tide (two times per day) was the same as seawater (~35 ppt), even in the upper area of the mangrove forest [19], suggesting that complete tidal flushing occurs within the small mangrove estuary.

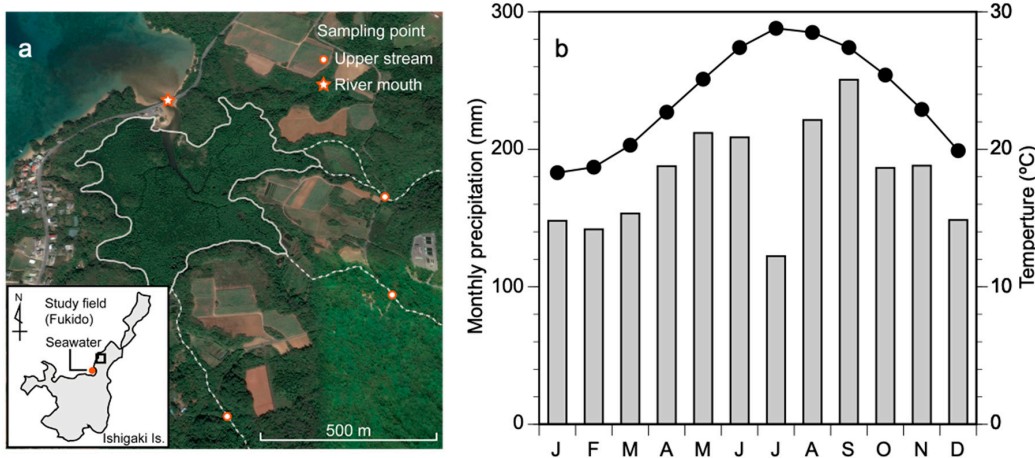

**Figure 1.** Site maps and sampling points of water (**a**). The white solid line indicates the mangrove estuary, while dotted lines indicate the inflowing upper streams. Map data provided by Google Earth. Monthly precipitation and monthly mean air temperature (•) near the study site (**b**). Data were obtained from a weather station (Ibaruma) located approximately 5 km from the study area.

## 2.2. Water Sampling and Analysis

Samples for pH, electric conductivity (EC), and concentrations of DIC and DOC analysis were collected at the river mouth from a depth of ~10 cm every hour over a 25 h time series in winter (8 and 9 March 2016) and summer (23 and 24 August 2016), respectively (Figure 1a). Water depth at the river mouth was ca. 50 cm during ebb tide, and the pH and EC did not differ depending on the depth. Samples of upper streams and seawater for the endmembers of the conservative mixing model were collected five times in the winter and three times in the summer during the time course (Figure 1a). To avoid the direct influence of rivers, we collected seawater samples approximately 2 km away from the river mouth. We measured pH using a portable meter (LAQUAtwinn series, HORIBA, Kyoto, Japan), with a reproducibility of ±0.01 pH units. EC measurements were conducted in situ using portable meters (LAQUAtwin series, HORIBA, Kyoto, Japan), with an accuracy of ±2%. Samples for DIC and DOC were collected with sample-rinsed polypropylene bottles, separately. The DIC concentration was measured in situ using a portable DIC measurement system, including an infrared gas analyzer (IRGA, LI-820, LI-COR, Lincoln, NE, USA). The DIC in the samples was vaporized to $CO_2$ by mixing with a sufficient amount of acid solution (0.1 M phosphoric acid). The resultant $CO_2$ was transported to the IRGA via an open flow system and quantified there. The DIC concentration of the sample water was expressed as mg C $L^{-1}$, considering the injected sample volume (1–2 mL). The coefficient of variation (CV) between four times measurements was 1.97%.

The samples for DOC were kept cool in the dark while being transported back to the laboratory and were filtered within 72 h of collection with a pre-combusted glass fiber filter (ADVANTEC GF-75). The carbon contamination from the filter was negligible (<0.01 mg C $L^{-1}$) according to this protocol. The DOC was measured as non-purgeable organic carbon in a total organic carbon analyzer (TOC-L$_{CPH}$, Shimadzu, Japan) using the platinum-catalyzed high-temperature combustion method coupled to non-dispersive infrared gas detection of $CO_2$ [22]. We sprayed the samples with $CO_2$-free carrier gas for 90 s in the built-in syringe of the TOC analyzer to remove any DIC prior to combustion after acidification. We then calibrated the samples by running four standards of potassium hydrogen phthalate solution over an appropriate range. Reported DOC concentrations are average values of triplicate measurements. The precision of measurements (CV) was higher than 2.0%.

Stable carbon isotope ratios of DIC samples during the time course of one tidal cycle (six samples, 2 h intervals) on March 8 were measured using a Thermo Fisher Scientific Model DELTA$^{plus}$ XL (SI Science Co. Ltd., Yokohama, Japan). Reproducibility of $\delta^{13}$C-DIC measurements was better than ±0.2‰. All stable carbon isotope ratios were calculated relative to the international standard (Pee Dee Belemnite).

### 2.3. Estimation of the Water Flux at the River Mouth

The amount of water flux flowing between the sea and mangrove area was measured using an acoustic Doppler current profiler (ADCP; SonTek IQ, YSI Yellow Springs, OH, USA; Nanotech Inc., Feldkirchen, Germany) during the period from 23 to 24 August 2016 with 5 min measuring intervals. To validate the measurement accuracy of the ADCP, flow velocities along the transect crossing near the river mouth were measured using an electromagnetic velocity meter (FLOMATE MODEL 2000, Marsh-McBirney Inc., Loveland, CO, USA). Water flux estimation during this period is required because it was not performed from 8 to 9 March 2016. For this purpose, we developed one index, which is analogous to Manning's equation commonly used for calculating river discharge. Here, we define this index as "flow index (*FI*)," expressed as follows:

$$FI = (TL^p - b)^{\frac{2}{3}}\left(TL^p - TL^{p-\Delta t}\right)^{\frac{1}{2}} \tag{1}$$

Here, *FI* is the water flow index ($m^{7/6}$); *b*, the lowest bottom elevation of the riverbed (m); *TL*, tidal level; and superscript *p* and $\Delta t$, the present time and time interval, respectively. Principally, the way to choose the duration of $\Delta t$ is arbitrary, and $\Delta t$ is related to the residential time within the mangrove area. For reference, Manning's equation for shallow water is shown as follows:

$$v = \frac{1}{n}h^{\frac{2}{3}}I^{\frac{1}{2}} \tag{2}$$

Here, *v* is the flow velocity (m s$^{-1}$); *h*, river depth (m); *I*, gradient of river water surface (-); and *n*, coefficient of roughness ($m^{2/3}$ s$^{-1}$). When comparing the two equations, the two terms on the right-hand side of Equation (1) can be considered to correspond to *h* and *I* of Equation (2). Since *n* and the short distance required to calculate *I* are fixed values, the *FI* value is an analogous expression of Manning's Equation.

Tidal level data is required to calculate *FI*. Tidal level with a time resolution of 1 h can be obtained from the Japan Meteorological Agency. Additionally, 1 h was chosen as $\Delta t$ in this study. The amount of water flux during the DIC and DOC measurements was estimated through the relationships between *FI* and observed water flux.

### 2.4. Estimation of DIC and DOC Flux from the Mangrove Estuary

Principally, the product of water flow volume and mangrove-derived DIC and DOC concentration should provide an estimate of DIC and DOC flux from the mangrove estuary alone. In the present study, a simple two-component conservative mixing model was applied [13].

$$f = (EC_{obs} - EC_R)/(EC_M - EC_R) \tag{3}$$

$$C_{mix} = f{\cdot}C_M + (1 - f){\cdot}C_R \tag{4}$$

Here, *f* is the seawater fraction at the river mouth; *EC*, electrical conductivity; and *C*, concentration of DIC and DOC. The subscripts *obs*, *mix*, *M*, and *R* denote the observed values, estimated values by conservative mixing, and fixed values for marine and riverine endmembers, respectively.

The added carbon ($\Delta C$, mg C L$^{-1}$) during transport along the mangrove estuary was calculated by considering the difference between observed ($C_{obs}$) and expected ($C_{mix}$) of DIC and DOC. This value can be expressed as follows:

$$\Delta C = (C_{obs} - C_{mix}) \tag{5}$$

Daily DIC and DOC fluxes (g C m$^{-2}$ d$^{-1}$) from the mangrove estuary accumulate the amount of water flux (m$^3$ s$^{-1}$) at the river mouth that flows to the sea during ebb tide and consecutive $\Delta C$ based on the water area covers in the mangrove estuary at high tide. The $\Delta C$ may include other C sources in addition to C from mangrove NPP. According to previous studies, the $\delta^{13}C$ values for added DOC

(source of DOC) using Keeling plots are similar to those of mangrove litter [10,13], whereas for DIC, potential sources include not only mineralization of residual organic matter in mangrove forests but also $CaCO_3$ dissolution [17]. However, Ho et al. [25] revealed that $CaCO_3$ dissolution was less than 10% of the lateral DIC export in mangrove-dominated rivers of the Florida Everglades. Hence, we assumed that $\Delta C$ is solely due to mangrove-derived C in this study.

## 3. Results

### 3.1. Changes in DIC and DOC Concentration with Tidal Amplitude

There was a clear diurnal trend with tidal amplitude in EC and pH (Figure 2a–d) in both the winter and summer, with increasing EC and pH in response to raised tidal height due to the dominance of seawater. The range of the EC in the winter (Figure 2c, 1.69–5.0 S m$^{-1}$) was more broad than in the summer (Figure 2d, 4.1–5.2 S m$^{-1}$) because the tidal level remained high and seawater predominated at the river mouth all day in the summer, although both days were close to the half-moon phase of the lunar cycle. The trend of DIC (Figure 2e,f) and DOC (Figure 2g,h) concentrations was opposite to the EC; the values were highest at ebb tide and decreased with increasing tidal levels with the lowest value at high tide. The DIC concentration in the winter (ranging from 21.8 to 30.3 mg C L$^{-1}$) was slightly lower than in the summer (23.6–5.6 mg C L$^{-1}$), partly due to reduced decomposition. The DOC concentrations in the winter (ranging from 0.88 to 1.48 mg C L$^{-1}$) were also slightly lower than in the summer (1.09–1.85 mg C L$^{-1}$). $\delta^{13}$C-DIC values also had a clear trend with tidal levels, ranging from −13.9‰ to −4.2‰ during a tidal cycle in the winter (Figure 3). The $\delta^{13}$C-DIC values were lower at ebb tide than at high tide.

### 3.2. Daily Flux of Dissolved Carbon (DIC and DOC) from Mangrove

The EC values of seawater in the winter and summer were similar (Table 1), and we used the highest EC value (not mean values) as the seawater endmember when calculating "*f*" due to the variations in seawater EC values (3.8 to 5.2 S m$^{-1}$). EC values of upper riverine water were usually less than 0.1 S m$^{-1}$. The mean DIC and DOC concentrations in seawater were higher than in upper riverine water during both the winter and summer.

**Table 1.** Marine and riverine conditions (EC and pH), dissolved inorganic (DIC) and organic carbon (DOC) concentrations in the winter (8 March) and summer (23 August)

| Sample | EC (S m$^{-1}$) | pH | DIC (mg C L$^{-1}$) | DOC (mg C L$^{-1}$) |
|---|---|---|---|---|
| **8 March 2016** | | | | |
| Marine | 4.63 ± 0.49 | 8.06 ± 0.42 | 24.7 ± 1.63 | 1.06 ± 0.22 |
| Riverine | 0.036 ± 0.010 | 7.85 ± 0.21 | 17.0 ± 0.91 | 0.88 ± 0.19 |
| **23 August 2016** | | | | |
| Marine | 4.77 ± 0.40 | 7.50 ± 0.77 | 28.3 ± 4.49 | 1.09 ± 0.09 |
| Riverine | 0.052 ± 0.020 | 7.24 ± 0.31 | 22.3 ± 2.83 | 1.04 ± 0.33 |

The DIC and DOC concentrations along the EC gradient at the river mouth displayed a marked non-conservative pattern with more scattering toward the freshwater during ebb tide (Figure 4). The DIC and DOC concentrations were all higher than those of the conservative mixing line ($\Delta C > 0$) during ebb tide, indicating carbon inputs from the mangrove forest. The $\Delta C$ of DIC and DOC at the river mouth was higher during ebb tide (low EC in Figure 4) than high tide (high EC in Figure 4). The $\Delta$DIC (mangrove-derived C) to total DIC concentration proportion in the winter and summer ranged from −10% to 34% (Figure 4a) and from −20% to 26% (Figure 4b), respectively. The $\Delta$DOC to total DOC concertation proportion in the winter and summer ranged from −16% to 56% (Figure 4c) and from −4% to 42% (Figure 4f), respectively.

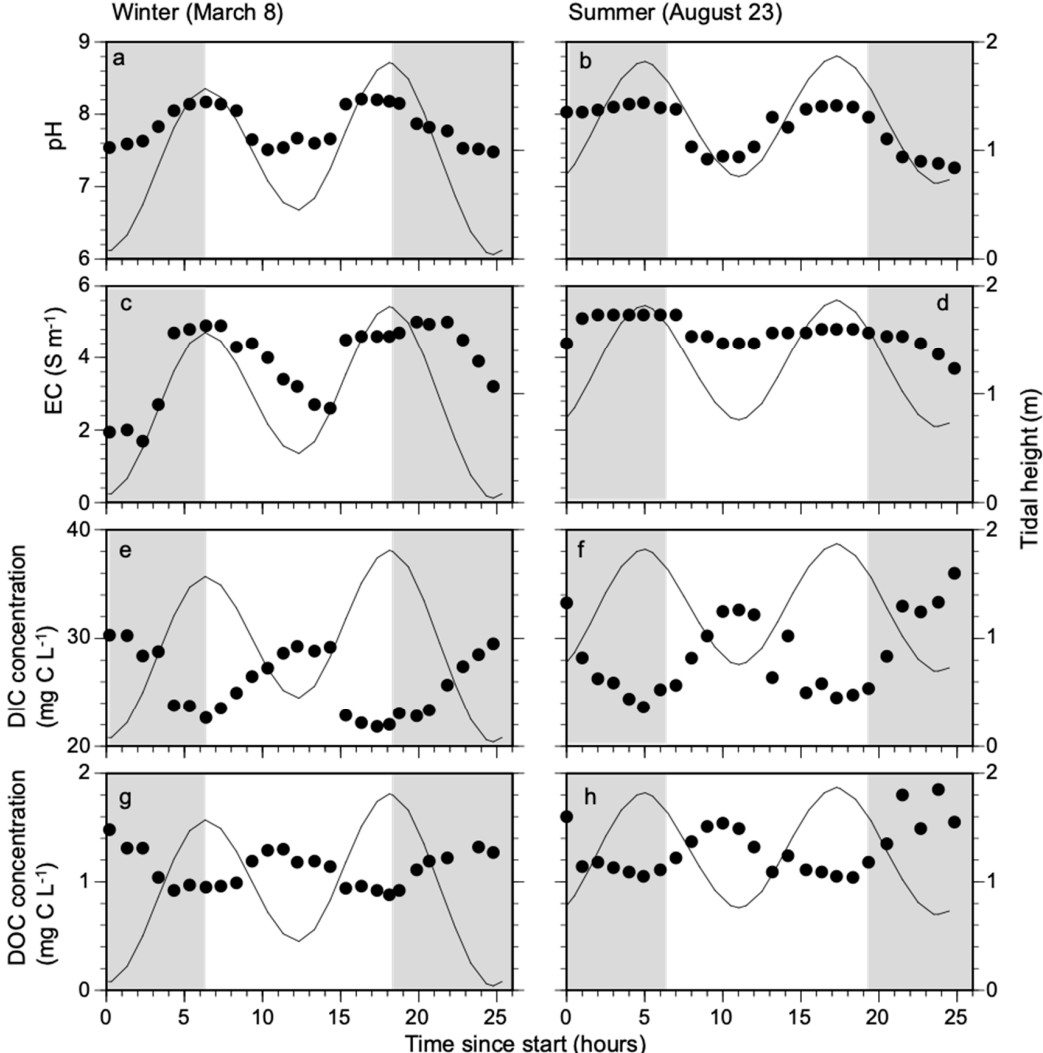

**Figure 2.** Tidal variations of pH (**a,b**), electric conductivity (EC) (**c,d**), dissolved inorganic carbon (DIC) (**e,f**), and dissolved organic carbon (DOC) (**g,h**) concentrations at the river mouth in a small estuarine mangrove during the time course. The line indicates the tidal height (m) obtained from the Japan Meteorological Agency (JMA). Shaded areas show the nighttime.

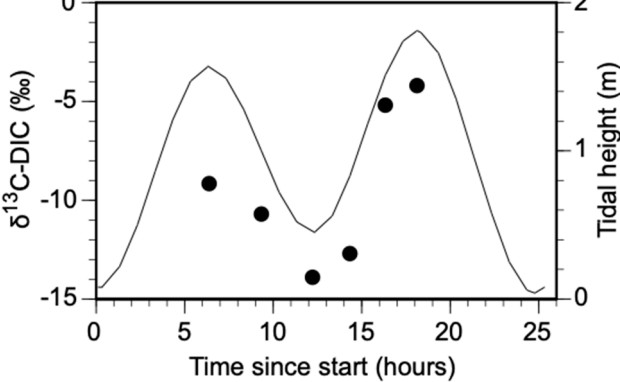

**Figure 3.** Variations of $\delta^{13}$C-DIC values at the river mouth during time course in winter. The line indicates the tidal height (m) obtained from the Japan Meteorological Agency (JMA).

The relationships between the estimated index of velocity (FI) and measured discharge by ADCP are shown in Figure 5. We established a very good site-specific relationship between FI and the amount of water flux ($Q\ \mathrm{m^3\ s^{-1}}$) as $Q = 18.214\ FI + 1.074$. There is no rational reason to choose other higher-order curves, although the fitted linear line shows several systematic errors. Hence, we chose a linear line for the first approximate estimate. The equation can be considered valid in terms of the whole range of tidal levels because these levels that were utilized to develop the Equation covered one neap tide to a spring tide. As the FI can only be calculated using tidal levels, water flux at any moment can only be estimated using the tidal levels at that moment and the hour immediately before the moment. In this way, we calculated the water flow volume during the period of the DIC measurements on 8 March 2016, and 23 August 2016. The direction of water flux at the river mouth sharply shifted from high tide to ebb tide (Figure 6a,b). The concentrations of ΔDIC and ΔDOC were highest during ebb tide, although the water fluxes at the river mouth were almost 0. The concentrations of ΔDIC (Figure 6c,d) and ΔDOC (Figure 6e,f) decreased to 0 with high tide and then recovered with ebb tide again. We summed the daily DIC and DOC fluxes when the discharge and ΔC > 0. The area-based DIC and DOC fluxes from the mangrove forest were estimated to be 1.36 and 0.20 $\mathrm{g\ C\ m^{-2}\ d^{-1}}$ in the winter and 3.35 and 0.86 $\mathrm{g\ C\ m^{-2}\ d^{-1}}$ in the summer, respectively.

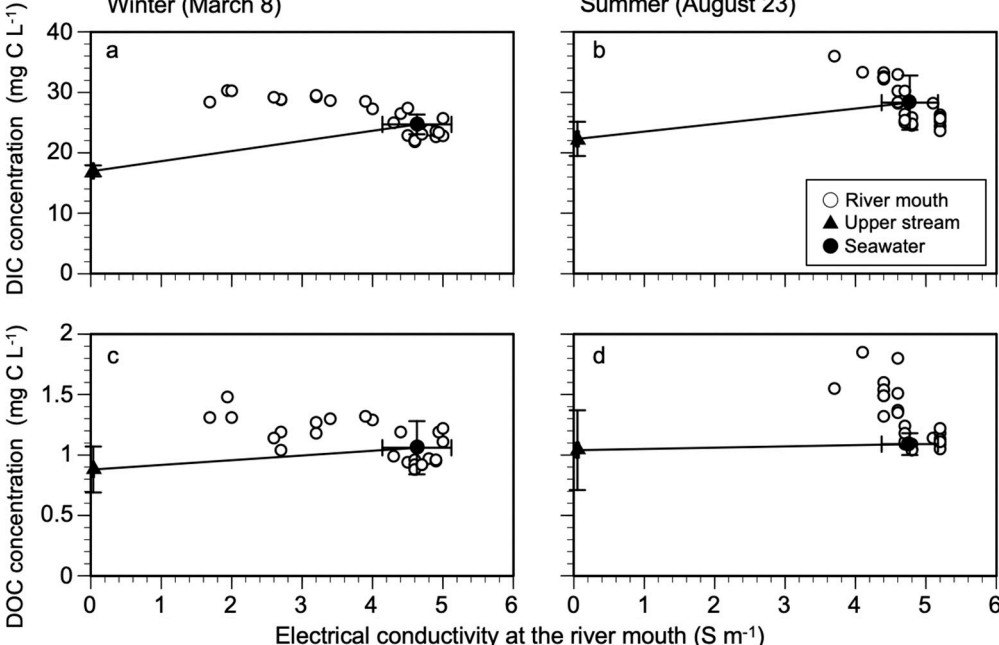

**Figure 4.** Relationship between EC and DIC (or DOC) concentrations at the river mouth water with tidal exchange (open circles) in a small estuarine mangrove in the winter (**a,c**) and summer (**b,d**). The line indicates the modeled carbon concentration ($C_{mix}$) derived by the conservative mixing of seawater (filled circle) and freshwater (filled triangle) with standard error (SE) as endmembers.

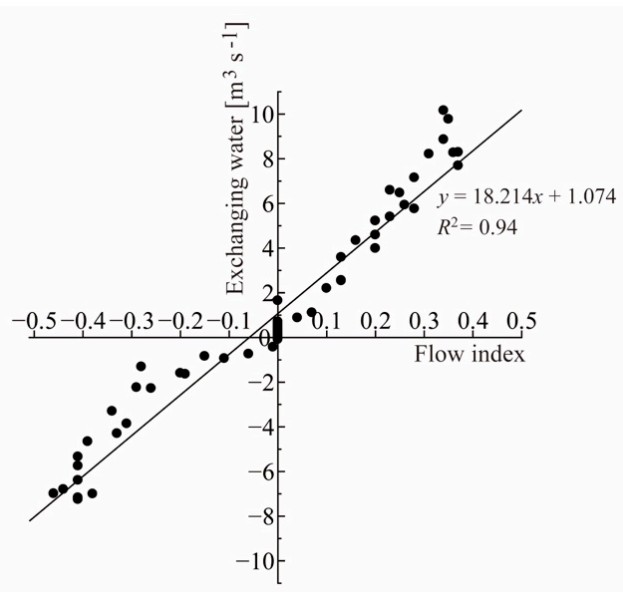

**Figure 5.** Relationships between the estimated index of velocity and measured discharge by ADCP at the river mouth in a small estuarine mangrove.

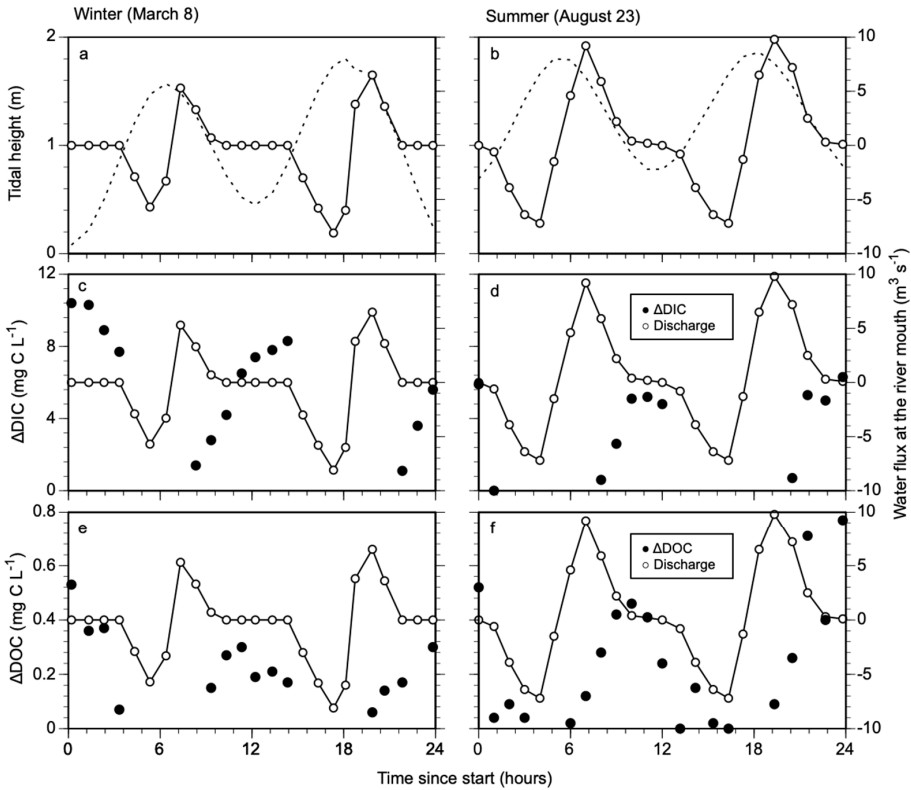

**Figure 6.** Tidal variations of water fluxes (**a**,**b**), ΔDIC (**c**,**d**), and ΔDOC (**e**,**f**) concentrations at the river mouth in a small estuarine mangrove during the time course. The broken line indicates the tidal height (m) obtained from the Japan Meteorological Agency. The solid line with open circles indicates the water fluxes (discharge) at the river mouth, the positive values denote the water flow from mangrove to the sea, and the negative values denote the water flow from the sea to the mangrove. The filled circles indicate ΔDIC and ΔDOC, which represent mangrove-derived carbon.

## 4. Discussion

### 4.1. Methods for Dissolved Carbon Export Estimates

Since Bouillon et al. [10] demonstrated that over 50% of mangrove productivity was unaccounted for in various carbon sinks, considerable efforts have been made to estimate lateral C fluxes from mangrove forests. However, few studies have attempted to investigate area-based fluxes to estimate mangrove NEP, partly due to methodological difficulty. There are two main methods for lateral C flux estimates from mangrove forests at the moment: the closed chamber method and the watershed-based method. Maher et al. [11] used sediment cores in intertidal areas within a mangrove forest to determine the diffusive DIC and DOC fluxes as a closed chamber method. They calculated mangrove area-based C flux using the functions of the concentration in the core and the volume of the overlying water. Li et al. [26] also used the closed chamber method and set them inside and outside mangrove forests to determine the concentration differences; this method was initially based on Dausse et al. [27]. However, DIC export is closely related to tidally driven porewater exchange from underground [10–12,28]. Thus, it may be challenging to evaluate the heterogeneity of underground structures, such as the distribution of roots and crab holes, using small cores with the closed chamber method.

By contrast, watershed-based estimates are another method to assess lateral C fluxes [13,18,29]. For instance, Call et al. [29] estimated the net exchange of DIC, DOC, and POC using the integration of discharge and consecutive concentrations, where the exchange rates were normalized to the catchment area of each creek and were estimated using Google Earth. However, they noted that the cross-sectional area and current velocity of large creeks were each assigned an uncertainty of 25%. Furthermore, the watershed area was not entirely covered by mangrove forests, and tidal amplitudes that influence the lateral C flux from mangrove forests greatly varies with their position (e.g., the position from the riverside and from the river mouth to the upper mangrove forests). Therefore, watershed-based estimates of lateral C fluxes are difficult to apply to mangrove area-based estimates of C budget.

In the present study, we estimated mangrove area-based lateral C fluxes using a small mangrove estuary that only had one exit for water exchange to the coast. We sampled the water at the mouth of the creek, which floods a known area of the estuarine mangrove, and integrated discharge and consecutive concentration of mangrove-derived DIC and DOC ($\Delta$C). Finally, we estimated the area-normalized C fluxes based on the area of inundated mangrove estuary. The daily fluxes based on our methods were mangrove area-based DIC and DOC fluxes that can be directly incorporated into the estimation of mangrove C budget. One uncertainty related to our model was the resolution of water sampling, especially during ebb tide. $\Delta$C decreased with increasing tidal levels (Figure 6) and was approximately 0 at the highest tide; then, the discharge direction at the river mouth abruptly shifted from negative (inflow) to positive (outflow). In this case, the slight change of $\Delta$C during the change of discharge direction is critical for our daily flux estimation. The most favorable period for aquatic C export would be during early ebb tide when C is flushed out of the system by litter leaching and porewater draining [30]. Therefore, we need a shorter time interval of water sampling, especially during the time of abrupt change for the discharge direction in the early ebb tide (Figure 6).

Another uncertainty related to our model was the assumption that the endmember (riverine and marine waters) was homogeneous in space, depth, and time. We quantified the $\Delta$C in the mangrove estuary (mangrove-derived C) using a two-component mixing model of freshwater and seawater (Figure 4). However, $\Delta$C sometimes displayed negative values (i.e., measured DIC and DOC were lower than the estimated mixing values) during high tide because EC and dissolved C concentrations of the endmembers had temporal and spatial variations (Table 1), although the negative values were within the range of the standard error (SE) of seawater-dissolved C concentrations (Figure 4).

A stable isotope mass balance model is also required to estimate the DIC and DOC source. As shown in Figure 3, preliminary data for $\delta^{13}$C-DIC varied during the tidal cycle in the Fukido mangrove, with the highest value (−4.2‰) during high tide and the lowest value (−13.9‰) during

low tide. The decreasing trend in $\delta^{13}$C-DIC during ebb is consistent with the tidal variations observed in other mangrove forests, e.g., the Bay of Bengal (range between −4.6‰ and −3.7‰) [13], Tanzania (−9.4‰ and +0.5‰) [10], Australia (−5.5‰ and +2.3‰) [11], and Vietnam (−12.6‰ and −8.6‰) [12]. The cause of the $^{13}$C depletion in low tide water may be dilution by a freshwater source or DIC inputs from mangrove forests, whereby the excess DIC shows a strongly negative $\delta^{13}$C-DIC signature similar to that of its source [10]. The role of carbonate mineral dissolution and precipitation on DIC concentrations might be important where the bedrock is composed of limestone, as in our study sites [17]; thus, further studies are required to examine the origin of DIC using isotopic signatures in more detail.

As for DOC, we previously investigated the dynamics of fluorescent dissolved organic matter (DOM) through excitation–emission matrix spectroscopy combined with parallel factor analysis (EEM-PARAFAC) for 2 years [22]. The EEM-derived biological index, an indicator of freshly produced DOM by microbes, lacked evidence of newly produced DOM inputs within the forest [22]. Water within the mangrove forest was more turbid than headwater and seawater, which precluded possible phytoplankton-derived DOC inputs within the forest.

### 4.2. Dissolved Carbon Flux with Tidal Amplitude

The DOC export rates in the Fukido mangrove in the winter and summer were 0.20 and 0.86 g C m$^{-2}$ d$^{-1}$, respectively. The DOC export was within the range previously reported for mangrove areas [26] of 0.56 to 7.10 Mg C ha$^{-1}$ year$^{-1}$ (0.15 to 1.9 g C m$^{-2}$ d$^{-1}$). Compared with organic carbon export (or import) from previous mangrove studies, there are few data on lateral DIC export concerning mangrove areas. Maher et al. [11] estimated that DIC export ranged from 2.2 to 4.1 g C m$^{-2}$ d$^{-1}$ on the east coast of Australia, which is the first estimation of mangrove area-based DIC exchange on a regional scale. More specifically, DIC outwelling rates of our study site (1.36 to 3.35 g C m$^{-2}$ d$^{-1}$) are within the range of other regional estimates, such as observed in an Australian mangrove forest (4.9 ± 2.4 g C m$^{-2}$ d$^{-1}$) [18], subtropical Taiwan mangrove forest (1.46 to 4.44 g C m$^{-2}$ d$^{-1}$, except for the heavily polluted river water) [26], and the Bay of Bengal (2.43 g C m$^{-2}$ d$^{-1}$) [13].

In contrast to the estimation of single daily fluxes of DIC and DOC (g m$^{-2}$ d$^{-1}$), spatial and temporal variations of daily fluxes represent another challenge to scaling up annual fluxes (Mg ha$^{-1}$ year$^{-1}$). Higher concentrations of dissolved C in mangrove forests are well reported during the wet season [12,28]. This result is usually explained by higher heterotrophic activity during the wet season due to intense rainfall, and organic matter is quickly mineralized. In our study sites, both daily DIC and DOC fluxes in the summer were more than two times higher than in the winter. ΔDIC and ΔDOC during ebb tide did not differ in such a way within the two seasons, whereas water discharge was higher in the summer due to higher tide levels (Figure 6). Therefore, not only the higher heterotrophic activity due to high temperature but also the seasonal change of water discharge resulted in higher dissolved C fluxes from the mangrove forest in the summer at our study site.

The tidal range of the lunar cycle also appeared to affect porewater input intensity because of the differences in volume and surface area of water that was in contact with the mangrove sediments at high tide [12,31]. However, there are few data available regarding variations with lunar cycles (spring to neap tide). Ray et al. [30] recently reported the mangrove-derived DOC and DIC exchange at the mouth of the Sinnamary estuary, French Guiana. DOC contributed the most to lateral C flux, whereas DIC was either exported or imported according to tide. Lateral DOC and DIC for a neap tide cycle during the dry season were 8.14 and −0.90 g C m$^{-2}$ d$^{-1}$, respectively. They concluded that the magnitude of C exchange fluxes was uncertain due to variability in tidal (spring vs. neap) and seasonal (wet vs. dry) patterns. There was no clear seasonal precipitation pattern in the mangrove area near the northern limit of mangrove distribution in East Asia, due to the monsoon climate (Figure 1b), compared with other tropical/subtropical mangrove areas. In this case, the lunar cycle would have a great effect on the magnitude of lateral C fluxes rather than seasonal temperature change in the present study site, although our daily flux estimates were only for the half-moon phase, possibly indicating

average C fluxes during the lunar cycle. Furthermore, Ishigaki Island is subjected to typhoons every year, and typhoons are another significant factor that increases water discharge in mangrove forests. Kida et al. [22] observed a sharp increase in DOC concentration due to heavy rain with typhoons in the Fukido River. More methods are required to estimate annual lateral fluxes while considering the lunar cycle and episodic events such as typhoons.

### 4.3. Mangrove Carbon Budget and Lateral C Fluxes

Webb et al. [32] reviewed the relative importance of the aquatic C flux in offsetting carbon uptake in various ecosystems. The relative magnitude of aquatic C flux offset varied widely across ecosystems, ranging from <1% in a boreal forest to 590% in a freshwater marsh, and highlights the uncertainty on the lateral C flux contribution to the net ecosystem carbon budget in mangrove forests. Furthermore, recent estimates of lateral C fluxes in mangrove forests suggest that the summation of exported DOC, DIC, and heterotrophic soil respiration greatly exceeded the amount of litter production [13,26]. The annual leaf litter production in the study site was 7.05 Mg dry matter $ha^{-1}$ $year^{-1}$ (0.97 g C $m^{-2}$ $d^{-1}$ if we assumed that the carbon content of dry weight was 50%) [19]. Thus, the outwelling of dissolved C from the mangrove estuary greatly exceeded leaf litter production. The export of mangrove-derived C exceeds the "missing carbon" and suggests that other unknown processes must balance the carbon budget. One possibility is that the additional C is primarily derived from outside of mangrove ecosystems (e.g., from the upper river or adjacent ecosystems). If outwelling C is derived from outside of mangrove forests, lateral C exports from the mangrove estuary do not directly contribute to the C budget of mangrove NEP. River water discharge in Fukido mangrove is less than 10% of tidal water exchange, and DOC concentrations of the upper Fukido River are low compared with the other mangrove forests [22]. However, we did not measure the exchange of POC or litter fluxes in the river. These results suggest that the lateral DIC and DOC fluxes come from mangrove sediments that originated in mangrove litter since the Fukido mangrove is a clearly defined estuary with tidal change (Figure 1).

Iimura et al. [24] studied the soil C stock and their primary origin in the Fukido mangrove. The soil C stock in the Fukido mangrove was 251 ± 35 Mg C $ha^{-1}$ at a depth of 90 cm, and dead fine roots in the soil, but not fallen litter, were significantly and positively related to the soil C stocks. Furthermore, most of the leaf litter may flow out from the Fukido mangrove toward the coast via tidal currents based on our observations [23]. Thus, an additional source of organic carbon, which did not account for the decomposition of litterfall in the sediments, was possibly derived from belowground detritus. Belowground NPP, especially fine root production, should make up a large component of total NPP in mangrove ecosystems. For example, the belowground NPP, including fine roots, was larger than the aboveground NPP in a secondary mangrove forest in eastern Thailand, although their estimates were only 30 cm in depth [33]. Robertson and Alongi [34] reported the first integrated estimates of the rate of turnover for fine root detritus in tropical Australian mangrove forests. The estimated fluxes of C via the decomposition of dead fine roots were very high, ranging from 2.5 to 18.8 g C $m^{-2}$ $d^{-1}$. Thus, vast fine root production and turnover within mangrove sediments may be one of the missing C sources that are not considered.

## 5. Conclusions

We estimated mangrove area-based DIC and DOC outwelling (lateral C fluxes from the mangrove forest) using the integrated discharge method and consecutive concentrations of mangrove-derived DIC and DOC ($\Delta C$) in the small mangrove estuary (Ishigaki Island, southwestern Japan). DIC and DOC concentrations at the river mouth increased during ebb tide in both summer and winter. The highest proportion of mangrove-derived C concentrations of DIC and DOC at the river mouth during ebb tide were between 34% and 56% in the winter and 26% and 42% in the summer, respectively. DIC and DOC fluxes from the mangrove estuary were estimated to be 1.36 g C $m^{-2}$ $d^{-1}$ and 0.20 g C $m^{-2}$ $d^{-1}$ in the winter and 3.35 g C $m^{-2}$ $d^{-1}$ and 0.86 g C $m^{-2}$ $d^{-1}$ in the summer, respectively. The outwelling of DIC from the mangrove estuary greatly exceeded aboveground litter production (0.97 g C $m^{-2}$ $d^{-1}$).

This result suggests that vast fine root detritus may balance the C budget in the mangrove forest. Still, little is known about the importance of dissolved C in the mangrove C budget, especially regarding subtropical mangrove forests, because mangrove forests in different environments may produce and store carbon in different ways [24]. We suggest further estimates of combined exports of POC, DOC, DIC, and $CO_2$ emissions from mangrove forests and the detection of seasonal and lunar cycle variations to have a complete understanding of the underlying mechanisms that make mangrove forests large carbon sinks.

**Author Contributions:** Conceptualization; T.O. (Toshiyuki Ohtsuka), N.F., Y.I., K.K.; methodology; T.O. (Toshiyuki Ohtsuka), S.Y., N.F., T.O. (Takeo Onishi); field inventory and water sampling; M.T., M.K. (Miyuki Kondo), T.O. (Takeo Onishi), M.K. (Morimaru Kida), V.S., R.C., Y.I., K.K.; laboratory analysis; S.Y., M.K. (M Kida), M.K. (Miyuki Kondo), M.T.; flux model and calculation; T.O. (Takeo Onishi); Writing; T.O. (Toshiyuki Ohtsuka), T.O. (Takeo Onishi), M.K. (Morimaru Kida); funding acquisition; T.O. (Toshiyuki Ohtsuka). All authors have read and agreed to the published version of the manuscript.

**Funding:** This study was supported by JSPS (Japan Society for the Promotion of Science) KAKENHI Grant Number 15K12186.

**Acknowledgments:** We thank the members of the Laboratory of Vegetation Function, River Basin Research Center (RBRC), Gifu University for their assistance with the field survey.

**Conflicts of Interest:** The authors declare no conflict of interest.

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
