# Peer review of "Lateral Export of Dissolved Inorganic and Organic Carbon from a Small Mangrove Estuary with Tidal Fluctuation"

_forests, doi:10.3390/f11101041_

Round 1

Reviewer 1 Report

Ohtsuka et al., write about dissolved phase C export from the riverine mangrove to the sea from a relatively mangroves of subtropical settings in southern Japan. Export and sequestration of ‘blue C’ have been a hot topic in current research on vegetated coastal systems. These aspects of marine C budget links to regional climate change and could extend to global basis. Therefore, more regional data, specially C export flux which is one of largest loss components in mangrove C budget, would enrich our understanding more.

Authors did well in making working strategy like fine scale time series involving seasonal contrasts. The method of calculating C flux using watershed discharge and non-conservative C fraction is well suited for this particular region. The writing is smart and straight forward. See also for reference (Sharma, S., Yasuoka, J., Nakamura, T., Watanabe, A., & Nadaoka, K. (2014). The role of hydroperiod, soil moisture and distance from the river mouth on soil organic matter in Fukido mangrove forest. In Proceeding of the Intenational Conference on Advances In Applied Science and Environmental Engineering (pp. 44-48).

However, I have several comments on this paper those should be addressed in the revised version.

  • Supportive evidence of DOC as entirely of mangrove origin is needed. The ‘added DOC’, according to author, was assumed to be purely of ‘mangrove’ origin, what was the basis of this assumption? Assuming no measured 13C-DOC data for this region, how about other proxies, like DOC:DON? Or optical properties? Also it is important to delineate mangrove-derived DOC, like litter leaching? pore water drainage? root exudates? or combinations of all these? Analytical measurement of DOC needs more details like standard used, accuracy and uncertainty in measurements. There was no account of DOC changes based on daylight and dark light conditions, similarly no explanation why ebb tide is more favorable for outwelling.

  • Authors should try keeling plots (see Bouillon work, Maher work) to determine the intercept 13C-DIC which would support in finding internally added DIC in the system. Other than limestone, mangroves are said to be the sources of DIC at the mouth. Be careful, it is mangrove-respired CO2 to be main source, not just ‘mangroves’. Give instrumental analytical precision for DIC, 13DIC and pH. Check good citation for the Bay of Bengal DIC outwelling. Is this data good for the the Bay of Bengal or French Guiana ? Because hile websearch, it appears the data of DIC export from French Guiana is different (Ray et al., 2020, JGR-BG, 125). Authors should compare their values with these settings and explain why such differences are. Aspects like geomorphic settings, hydrologic regime, mangrove biomass are the key points behind the continental changes in C outwelling (both DOC and DIC)

  • In the model, one caveat lying within surface water sampling only, while bottom water could behave differently in the mixing model. This should be stated along with the importance of multisite time series observation to define site-specific flux variability within short distances of study site. Fine root production has been identified as unaccounted C, rightly so when we know how important fine root production is as C sink, bit then what about the bioturbation activities that might also contribute to the missing C via sinking litter in soil pockets by the crabs?

  • Argument of NEP was mostly speculative without any observed data support from current study instead previous works and that too might have differed from the time frame of this study. For example, soil respiration in mangrove can change dramatically at interannual basis, if so, then data comparison b/w different time frames may not make any sense. Nevertheless, the strong point of this paper is the export rate calculation and that aspect has be given exclusively in the title, omitting NEP (description in the discussion should be OK)

  • It is a bit hard to understand why the authors needed to develop FI instead of just using the water flux from the ADCP data. (also, the data from ADCP is just a current profile at one point, therefore a description on how to calculate water flux from ADCP data may be needed.). The "I" in Manning's equation is water surface slope (dh/dx), and this is essentially different from the FI which uses (dh/dt) for "I". (h: water depth, x: distance, t: time). Even if there is an empirical relationship between water flux and FI (which is clear from the results), I think the relationship is specific to the site only, and cannot be applied to other sites. More clarification is needed here.

  • Also there are strong relationships between the tidal fluctuations and the water flux patterns because Flow Index was determined only by tides. The Flow Index would be valid if the data for the comparison covers one tidal period from neap tide to spring tide. But that was not clarified here.

  • Please insert legends in the plots where secondary y-axis are given, otherwise its difficult to understand easily.

Reviewer 2 Report

The manuscript describes field measurements and calculation of lateral carbon flux in a mangrove estuary with a single outlet. The manuscript is written clearly with appropriately detailed descriptions of sampling, analysis, and calculations. The objective of the work was to test the hypothesis that carbon fluxes with tidal exchange contribute substantially to the overall carbon budget. While the study successfully quantified daily lateral carbon flux, the carbon budget for the mangrove estuary was not clearly presented. C stock and flux estimates for a variety of ecosystem processes are discussed, but there is not adequate context for the magnitude of the lateral carbon flux. To address this concern, I would highly suggest that the authors present a site conceptual model and associated data table that shows the complete set of known C pools and fluxes. Without this presentation, it’s challenging to assess the importance of the lateral carbon flux and truly address the stated objective of the manuscript. This specifically should be done in section 4.3 Mangrove carbon budget and NEP, which does not currently provide and NEP estimate.  If an estimate of NEP is not possible due to other major gaps in the carbon budget, then I would recommend recasting the objective and title the manuscript.

Line-by-line comments:

237: Table 1 formatting cuts off the caption.

239-242: Figure 4 caption should provide clearer description of which point is the freshwater and which the seawater endmember.

257-258: Despite the strong linear fit, it appears there are still systematic errors (e.g. residuals are not evenly distributed around the fit line). If the relationship does not fit normal assumptions (this should be evaluated), then other models could be constructed. This would matter for estimating the exchanging water flux, particularly at flow index values near 0.

342-344: it’s not clear what this sentence is trying to say, specifically the “rather than…” clause. What temperature effects would be expected, and how would mineralization change the observed flux dynamics?

353: This section should present as close to a complete NEP budget as possible. Blank line items are okay – they simply suggest the necessity of future research. Most of the discussion in this section does not read clearly because it is direct and how to create an NEP budget without actually performing the work. It would be a lot stronger to list all the appropriate stocks and fluxes in a table, which would also emphasize the magnitude of the lateral carbon flux relative to both the stocks of C in the system and the other fluxes.

354-356: this sentence does not read clearly and requires revision for clarity.

392: The calculations for these percentages should be included in the results or in earlier discussion; they appear for the first time in the conclusion section (other than the abstract). It would be very helpful to see the full carbon flux estimate and apportionment into mangrove-derived and non-mangrove-derived in a separate table. Are there error estimates for these values?

Reviewer 3 Report

This manuscript is generally well written. The contribution of lateral carbon fluxes to the carbon budget of mangroves has been ignored and needs to be reexamined. This study can add a case study showing its significant contribution. The methods of estimating lateral carbon fluxes were adequately described and the results were clearly presented. However, I have some concerns mainly about the background information on the study site, the limitation of the application of the study methods, and the lunar cycles of research design.

  1. The tree density and the age of studied forests are suggested to supplement for the background information.
  2. L120-122 There are two mangrove species in the estuary. The aboveground biomass and NPP of the mangrove provided in the text were for the two species? please clarify.
  3. L121 The word "mangroves" is often used to indicate "mangrove forest".
  4. L132 Was the 2169 mm of annual precipitation distributed evenly throughout the year?
  5. L142 Additional samples were collected for what?
  6. L199 any reference to assume that other C sources such as algal excretion and CaCO3 dissolution from the estuary were minor.
  7. Fig. 2a-d "There was a clear diurnal trend..." I cannot see when was the daytime from the figures.
  8. L207 both days were close to the half-moon phase. Why did the authors choose the half-moon phase? Why not the full-moon phase?
  9. Please rephrase the sentences L231~234. The link between the broader EC range and more seawater delivery needs more elaboration.
  10. Are the methods of this study limited only to the application of the mangroves with only one exit for water exchange to the coast? 

Round 2

Reviewer 1 Report

Authors have improved the manuscript following reviewers  comments.